# Measurement of average decoding rates of the 61 sense codons in vivo

**Justin Gardin[1†], Rukhsana Yeasmin[2†], Alisa Yurovsky[1], Ying Cai[1], Steve Skiena[2], Bruce Futcher[1]\***

[1]Department of Molecular Genetics and Microbiology, Stony Brook University, Stony Brook, United States; [2]Department of Computer Science, Stony Brook University, Stony Brook, United States

**Abstract** Most amino acids can be encoded by several synonymous codons, which are used at unequal frequencies. The significance of unequal codon usage remains unclear. One hypothesis is that frequent codons are translated relatively rapidly. However, there is little direct, in vivo, evidence regarding codon-specific translation rates. In this study, we generate high-coverage data using ribosome profiling in yeast, analyze using a novel algorithm, and deduce events at the A- and P-sites of the ribosome. Different codons are decoded at different rates in the A-site. In general, frequent codons are decoded more quickly than rare codons, and AT-rich codons are decoded more quickly than GC-rich codons. At the P-site, proline is slow in forming peptide bonds. We also apply our algorithm to short footprints from a different conformation of the ribosome and find strong amino acid-specific (not codon-specific) effects that may reflect interactions with the exit tunnel of the ribosome.

**\*For correspondence:** bfutcher@gmail.com

†These authors contributed equally to this work

**Competing interests:** The authors declare that no competing interests exist.

**Reviewing editor**: Nahum Sonenberg, McGill University, Canada

## Introduction

Different synonymous codons are used in genes at very different frequencies, and the reasons for this biased codon usage have been debated for three decades (*Fitch, 1976*; *Hasegawa et al., 1979*; *Miyata et al., 1979*; *Bennetzen and Hall, 1982*; *Lipman and Wilbur, 1983*; *Sharp and Li, 1986*; *Bulmer, 1987*; *Drummond and Wilke, 2008*) (reviewed by *Plotkin and Kudla (2011)*; *Forster (2012)*; *Novoa and Ribas de Pouplana (2012)*). In particular, it has been suggested that the frequently-used codons are translated more rapidly than rarely-used codons, perhaps because tRNAs for the frequent codons are relatively highly expressed (*Plotkin and Kudla, 2011*). However, there have also been competing hypotheses, including the idea that frequently-used codons are translated more accurately (*Plotkin and Kudla, 2011*). Genes are often recoded to use frequent codons to increase protein expression (*Burgess-Brown et al., 2008*; *Maertens et al., 2010*), but without any solid understanding of why this manipulation is effective. There is little or no direct in vivo evidence as to whether the more common codons are indeed translated more rapidly than the rarer codons. Even if they are, the fact that translation is typically limited by initiation, not elongation, leaves the effectiveness of codon optimization a puzzle (*Plotkin and Kudla, 2011*).

Ribosome profiling (*Ingolia et al., 2009*) allows the observation of positions of ribosomes on translating cellular mRNAs. The basis of the method is that a translating ribosome protects a region of mRNA from nuclease digestion, generating a 30 base 'footprint'. The footprint is roughly centered on the A-site of the ribosome. If some particular codon in the A-site were translated slowly, then the ribosome would dwell at this position, and so footprints generated from ribosomes at this position would be relatively common. Thus, if one looked at the number of ribosome footprints generated along an mRNA, there should be more footprints centered at every codon that is translated slowly and fewer centered at every codon translated rapidly; in principle, this is a method for measuring rates of translation of individual codons.

**eLife digest** Genes contain the instructions for making proteins from molecules called amino acids. These instructions are encoded in the order of the four building blocks that make up DNA, which are symbolized by the letters A, T, C, and G. The DNA of a gene is first copied to make a molecule of RNA, and then the letters in the RNA are read in groups of three (called 'codons') by a cellular machine called a ribosome. 'Sense codons' each specify one amino acid, and the ribosome decodes hundreds or thousands of these codons into a chain of amino acids to form a protein. 'Stop codons' do not encode amino acids but instead instruct the ribosome to stop building a protein when the chain is completed.

Most proteins are built from 20 different kinds of amino acid, but there are 61 sense codons. As such, up to six codons can code for the same amino acid. The multiple codons for a single amino acid, however, are not used equally in gene sequences—some are used much more often than others.

Now, Gardin, Yeasmin et al. have instantly halted the on-going processes of decoding genes and building proteins in yeast cells. Codons being translated into amino acids are trapped inside the ribosome; and codons that take the longest to decode are trapped most often. By using a computer algorithm, Gardin, Yeasmin et al. were able to measure just how often each kind of sense codon was trapped inside the ribosome and use this as a measure of how quickly each codon is decoded. The more often a given codon is used in a gene sequence, the less likely it was found to be trapped inside the ribosome—which suggests that these codons are decoded quicker than other codons and pass through the ribosome more quickly. Put another way, it appears that genes tend to use the codons that can be read the fastest.

Certain properties of a codon also affected its decoding speed. Codons with more As and Ts, for example, are decoded faster than codons with more Cs and Gs. Furthermore, whenever a chemically unusual amino acid called proline has to be added to a new protein chain, it slowed down the speed at which the protein was built. The method described by Gardin, Yeasmin et al. for peering into a decoding ribosome may now help future studies that aim to answer other questions about how proteins are built.

Experimentally, there is dramatic variation in the number of footprints generated at different positions along any particular mRNA (*Ingolia et al., 2011*) (*Figure 1*). However, these large peaks and valleys do not correlate with particular codons (*Ingolia et al., 2011*; *Charneski and Hurst, 2013*). It is still unclear what features of the mRNA cause the peaks and valleys, though there is evidence that prolines, or a poly-basic amino acid stretch, contribute to a slowing of the ribosome and a peak of ribosome footprints (*Ingolia et al., 2011*; *Brandman et al., 2012*; *Charneski and Hurst, 2013*).

Still, the fact that prolines and poly-basic amino acid stretches affect translation speed does not tell us whether different synonymous codons may also cause smaller effects. This question was investigated by *Qian et al. (2012)* and *Charneski and Hurst (2013)* using the yeast ribosome profiling data of *Ingolia et al. (2009)*. Neither group found any effect of different synonymous codons on translation rate—that is, perhaps surprisingly, each codon, rare or common, appeared to be translated at the same rate (*Qian et al., 2012*; *Charneski and Hurst, 2013*).

We have re-investigated this issue with two differences from these previous investigations. First, we have generated four yeast ribosome profiling datasets by optimized methods, including the flash-freezing of growing cells before the addition of cycloheximide ('Materials and methods'); Ingolia et al. added cycloheximide before harvesting cells. Second, we have developed a novel method of analysis, designed with the knowledge that, at best, codon decoding rates could account for only a small portion of the variation in ribosome footprints across an mRNA ('Materials and methods'). The combination of optimized data and novel analysis reveals that different codons are decoded at different rates.

## Results

In principle, using the ribosome footprint data to establish occupancy as a function of position might seem easy: align the reads to the reference genome to identify the 10 or so codons under each read, and tabulate the frequency of each codon observed in each position. Analysis of this general kind has

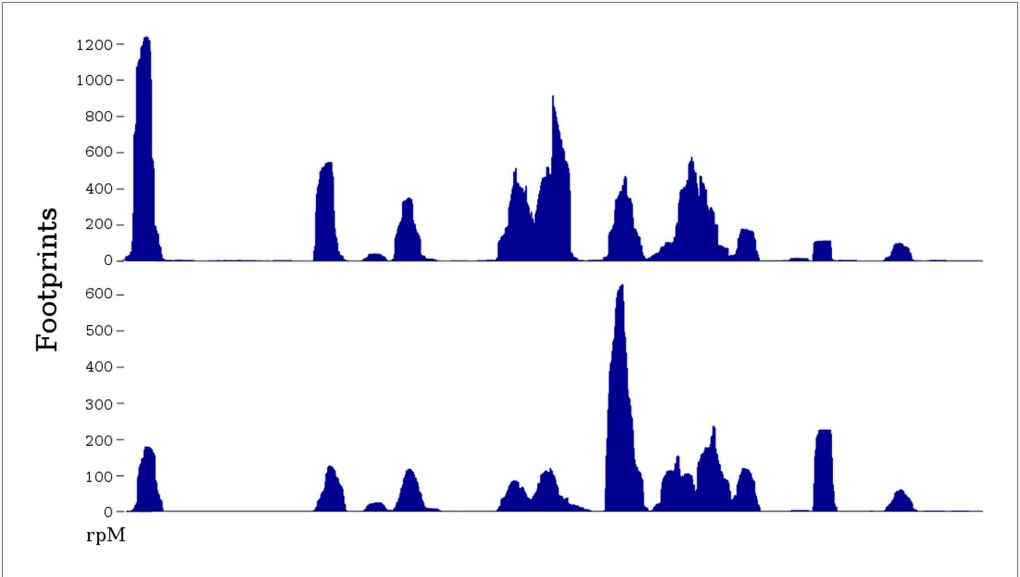

**Figure 1**. Two ribosome profiles of the *TDH1* gene. Top profile is from the data of **Ingolia et al., 2009**; bottom profile is from the SC-lys dataset ('Materials and methods'). The first (leftmost) peak in the profiles is at the ATG start codon; it may differ in relative height because the SC-lys dataset was generated using flash-freezing.

been carried out previously, but without detecting codon-specific differences in decoding rates (**Qian et al., 2012**; **Charneski and Hurst, 2013**). However, this analysis in its simplest form would overweight the highly expressed genes, which account for a large fraction of total reads—that is, a relatively small number of highly expressed genes would dominate the analysis. But because there are extreme peaks and valleys in ribosome footprint profiles (**Figure 1**), and because these are not primarily due to codon usage, this simple analysis would likely fail, because the results would depend mainly on a relatively small number of chromosome positions, and because of the peak-to-valley variability affecting these positions. Defining the right normalizations to compensate for differences in gene expression, gene length, sequence composition, etc, is complicated and problematic.

Instead, we have opted for a simpler approach. We independently analyze many selected regions (windows) where the effects of codon usage are particularly easy to assay. For each codon, we identify all translated regions in the genome where a particular codon (say CTC) occurs uniquely within a window of 10 codons upstream and 10 codons downstream—that is, a window 19-codons wide, with the codon of interest occurring exactly once at position 10 of the 19-position window. For footprints 10-codons long, there are exactly 10 classes of footprints that contain this particular CTC and fit entirely in the window. That is, the CTC of interest can occur at position 1 of the footprint, or position 2, …., or position 10. Analysis was restricted to windows with at least 20 total reads and at least 3 non-empty classes. For our four datasets discussed below, there was an average of 408, 1586, 1749, or 2868 qualifying windows per codon, respectively (more windows for the abundant codons, fewer for the rare codons).

In the absence of any codon preference of the ribosome, there should be a uniform distribution of footprints across the ten positions. That is, in a window centered on CTC and containing 100 footprints, one expects 10 footprints at each of the 10 positions, a relative frequency of 0.1 (10/100) at each position. On the other hand, if the ribosome was to dwell for an extended time over the CTC whenever that codon was at, say, position 6 of the footprint, then there might be 30 footprints with CTC in position 6, and about 8 footprints at each of the other 9 positions, thus giving a frequency distribution with a peak at position 6. Many such relative frequency distributions can be fairly averaged over all windows over all genes centered on a specific codon. Regions on highly expressed genes can be fairly compared with similar regions on genes with lower expression, because we are dealing with relative frequency distributions. Each window thus represents an independent trial of the ribosome's dwell time over each given codon. Averaging over the hundreds or thousands of windows in the genome generates a statistically rigorous analysis. Note that we do not attempt any normalization

based on gene expression—instead, we take each qualifying window as an independent experiment, regardless of level of expression, then average all frequency distributions from all windows for each codon. A related idea was also used by *Lareau et al. (2014)*, although on significantly different data, and with normalization by gene.

The relative frequency averaged over all windows is a number between 0 and 1, and we compare this to the baseline frequency (0.1) (total footprints over 10 positions) to compute a final statistic, which we call the Ribosome Residence Time, or RRT. For instance, if the average relative frequency for a codon at a particular position is 0.1, then the RRT is 1, and we interpret this to mean that the ribosome spends the average amount of time at the given codon at the given position. An RRT of two suggests that the ribosome spends twice as long as average at the given codon.

## Validation of ribosome residence time analysis

We tested this method of analysis using simulated and real positive and negative control data. For a simulated negative control, we assigned real footprint data from our SC-lys dataset to random codons and did RRT analysis. As expected, all codons at all positions show an RRT of about 1, that is, no signal (*Figure 2A*). For a simulated positive control, we generated a simulated data set of 2 million 10-codon reads over coding genes, but we biased these simulated reads to give more reads for the codon AAA at position 6 of the footprint. As expected, RRT analysis shows a peak for AAA at position 6 (*Figure 2B*).

For a real-data negative control, we pooled the control mRNA-seq data for 30 bp fragments from our four experiments ('Materials and methods') and analyzed these mRNA fragments. Since this RNA came from a total naked RNA preparation, there were no ribosomes and no ribosome footprints, so there should not be any signal from translation, even though we are analyzing real 30 bp RNA fragments. Indeed, RRT analysis shows no peaks in positions 2 through 9 of these fragments (*Figure 2C*). However, there are modest deviations from 1 at the termini, positions 1 and 10. We attribute these to some base-specificity for the enzymatic reactions used to generate the fragment library (*Lamm et al., 2011*;

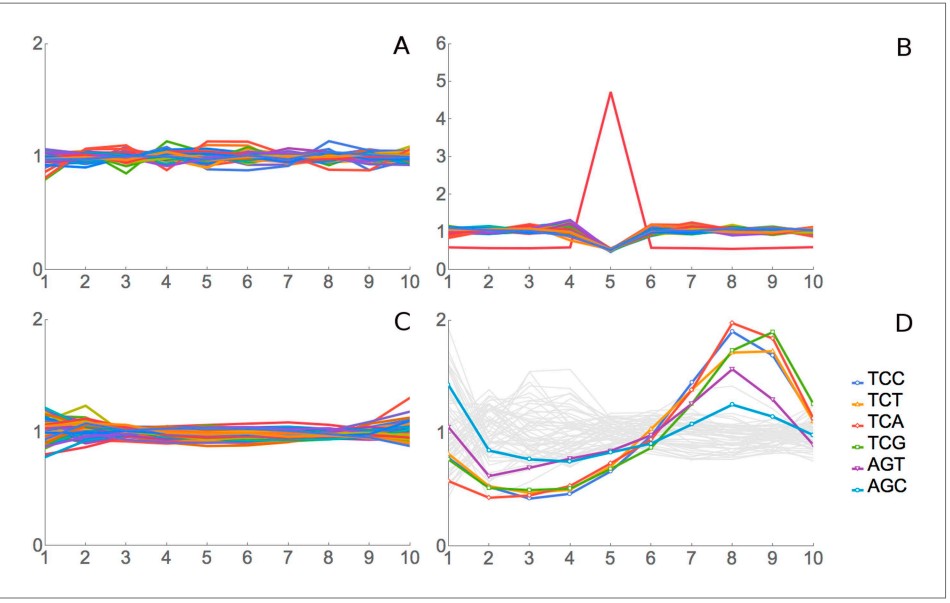

**Figure 2**. Validation for ribosome residence time analysis. (**A**) Simulated data, negative control. Real footprint data from the SC-lys dataset were randomly assigned to codons, and RRT analysis was carried out. A flat line with an RRT value of 1 indicates no signal. (**B**) Simulated data, positive control. A dataset of 2 million simulated reads was generated but biased to give more reads over the codon AAA at position 6. (**C**) Real data, negative control. RNA-seq data from naked fragments of RNA 30 nucleotides long, processed as if for ribosome profiling, were analyzed. (**D**) Real data, positive control. Real ribosome footprinting data from Li et al. were analyzed (*Li et al., 2012*). In this experiment, *E. coli* were starved for serine. Note that the highest Ser peak is for TCA, which is the rarest Ser codon in *E. coli*, and the lowest Ser peak is for AGC, which is the most common Ser codon in *E. coli*. High values at position 9 as well as 8 may indicate that the A-site may be at position 8 in some fragments and position 9 in others.

*Jackson et al., 2014*; *Raabe et al., 2014*). Supporting this interpretation, the same peaks and valleys at positions 1 and 10 (i.e., the same base-specificity) were seen in real ribosome-footprint data (see below).

For a real data positive control experiment, we used the *Escherichia coli* data generated by Li et al., who starved *E. coli* for serine, and did ribosome profiling (*Li et al., 2012*). Because of the starvation for serine, there is an expectation that all six serine codons should be decoded slowly and so should have high RRT values. This proved to be the case (*Figure 2D*). The six serine codons had 6 of the 7 highest RRT values at position 8 (*Figure 2D*, *Table 1*), which presumably represents the A-site in this experiment. Note that because these are *E. coli* ribosomes, the phase of the footprint (i.e., the position of the A-site in the footprint) is different from its phase with regard to yeast ribosomes (see below). The RRT analysis of *E. coli* footprints also showed interesting variation at positions 2, 3, and 4 (*Figure 2D*), which we will consider elsewhere.

*Lareau et al. (2014)* starved *Saccharomyces cerevisiae* for histidine using the His3 inhibitor 3-aminotriazole. This was another potential positive control, where the two His codons should be decoded slowly. We analyzed these ribosome profiling data. However, of the 11 million reads obtained in that experiment, about 10.6 million mapped to ribosomal RNA. The remaining ~0.4 million reads mapped to mRNA, but gave only 10 (ten) total windows passing our quality filters for RRT analysis, and this is too few. However, when we relaxed the filters to obtain more (albeit lower quality) windows, we observed obvious peaks (high RRT values) for both histidine codons at position 6 specifically in the 3-aminotriazole experiment (data not shown).

## Ribosome residence time analysis of codons

Having found that RRT analysis gives the expected results in control experiments, we applied it to the analysis of four of our ribosome profiling experiments. Our experiments differ from those of Ingolia et al. and Lareau et al., in that in those studies, cycloheximide was added to the growing yeast culture before harvesting (*Ingolia et al., 2009*; *Lareau et al., 2014*), whereas we harvest by flash-freezing and later add cycloheximide to the frozen cells ('Materials and methods'). The nature of our results is shown in *Figure 3* using the rare Leu codon CTC as an example. In this example, 10 codon (30 nucleotide) footprints that have CTC as the first codon have about the average relative frequency—that is, they have about the same relative frequency as footprints with any other codon at the first position. Similarly when CTC is in the 2nd, 3rd, 4th, 7th, 8th, 9th, and 10th positions. However, there is a relative over abundance of footprints that have CTC at the 6th position. In fact, for CTC at the 6th position, averaged over 451 windows (in the case of this rare codon), there are 1.89-fold more footprints than at the baseline. This suggests that ribosomes move relatively slowly when CTC is at the 6th position, and, therefore, these ribosomes are more frequently captured as footprints. We say that CTC has a Ribosome Residence Time (RRT) of 1.89 at position 6.

*Figure 4* shows data for all 61 sense codons from one of four experiments, the 'SC-lys' experiment. In a large majority of cases, a codon has its highest or lowest footprint abundance when the codon is in position 6. We interpret this to mean that the codon affects the rate of ribosome movement when

**Table 1.** Top ten RRTs at position 8 in *E. coli* starved for serine

| Codon | AA | Usage | RRT |
| --- | --- | --- | --- |
| TCA | Ser | 8.1 | 1.98 |
| TCC | Ser | 9.0 | 1.90 |
| TCG | Ser | 8.8 | 1.73 |
| TCT | Ser | 8.7 | 1.71 |
| AGT | Ser | 9.4 | 1.57 |
| ATA | Ile | 5.5 | 1.42 |
| AGC | Ser | 16.0 | 1.25 |
| ATT | Ile | 29.7 | 1.18 |
| CCT | Pro | 7.2 | 1.15 |
| CCA | Pro | 8.4 | 1.13 |

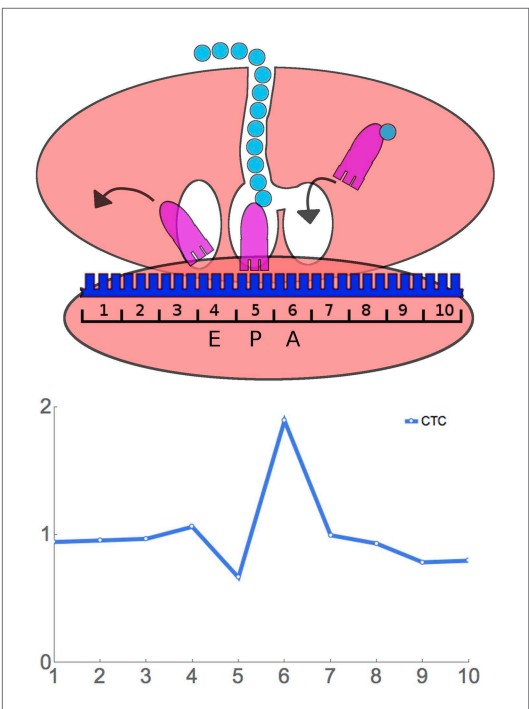

**Figure 3**. Principle of ribosome residence time analysis. The ribosome protects a 30 nt 'footprint' of RNA centered around the A, P, and E sites (positions 6, 5, and 4). The rare Leu codon CTC has a high RRT at position 6, which is likely the A-site.

the codon is in position 6, which we believe to be the A-site of the ribosome (see below for further support for this assignment). The behavior of the six Leu codons and the four Thr codons is highlighted in *Figure 4B,C*. Footprint frequencies also differ from the average in a specific way at positions 5 (*Figure 4D*) (see below) and 1 and 10, the two ends of the footprint. We attribute variation at positions 1 and 10 to some base-specificity for the enzymatic reactions involved in generating and analyzing ribosome footprints (*Lamm et al., 2011*; *Jackson et al., 2014*; *Raabe et al., 2014*); the same variations are seen in reactions with naked RNA fragments.

*Figure 5A* shows the deduced rate of ribosome movement for each codon, plotted against the frequency of codon usage. There is a good correlation (r = −0.52); that is, the ribosome moves faster over the more common codons.

There is also a correlation, albeit weaker, with the AT-richness of the codon. AT-rich codons are decoded somewhat faster than average, while GC-rich codons are decoded more slowly (*Figure 5B*). The mean RRT of codons with 3 or 2 GC residues was 1.23, while the mean RRT of codons with 1 or 0 GC residues was 1.01, a statistically significant difference (p < 0.003 by a two-tailed *t* test).

*Table 2* shows the Ribosome Residence Time at position 6 for each of the 61 sense codons. The slowest codon is the rare Leu codon CTC. Relatively, the ribosome spends about 1.9 times as long with a CTC codon in the A site as it does at the average codon. If the yeast ribosome spends 50 milliseconds (*Futcher et al., 1999*) on an average codon in the A-site, then the RRT suggests it spends about 95 milliseconds on CTC codons. The fastest codon is the relatively abundant Thr codon ACC (*Figure 4C*, *Table 2*), where it spends 0.70 times as long as average (i.e., about 35 milliseconds).

There are also peaks at position 5 (*Figure 4A,D*), which we interpret as the ribosome's P-site, where the peptide bond is formed. All four Pro codons are high at position 5: CCT, CCA, and CCC are the three slowest codons at position 5, while CCG is 6th (*Figure 4D*, *Table 2*). Proline is a unique amino acid in having a secondary rather than a primary amino group, and so it is less reactive in peptide bond formation. Proline forms peptide bonds slowly (*Muto and Ito, 2008*; *Wohlgemuth et al., 2008*; *Pavlov et al., 2009*; *Johansson et al., 2011*), and proline has been associated with slow translation in footprinting experiments (*Ingolia et al., 2011*). Our result that the ribosome slows with proline at position 5 is consistent with this and tends to confirm our assignment of position 5 to the P-site and, therefore, position 6 to the A-site. A few other residues also seem slightly slow at position 5 (e.g., Asn, Gly, see *Table 2* and *Supplementary file 1*), possibly due to low reactivity in peptide bond formation (*Johansson et al., 2011*).

All four proline codons also have high RRTs at position 6, the A-site (*Figure 4D*, *Table 2*). The dipeptide ProPro is translated very slowly (*Doerfel et al., 2013*; *Gutierrez et al., 2013*; *Peil et al., 2013*; *Ude et al., 2013*). We wondered whether the apparent slowness of proline at both positions 5 and 6 was an informatic artefact due to extreme slowness for ProPro dipeptides. We redid the original analysis after excluding all footprints encoding ProPro dipeptides. Results did not change significantly; Pro still appeared to be slow at both positions 5 and 6 (*Figure 6A*). On the other hand, when we looked specifically at footprints containing a ProPro dipeptide, there was a very large peak at position 5 (*Figure 6B*), consistent with the very slow peptide bond formation seen in studies cited above.

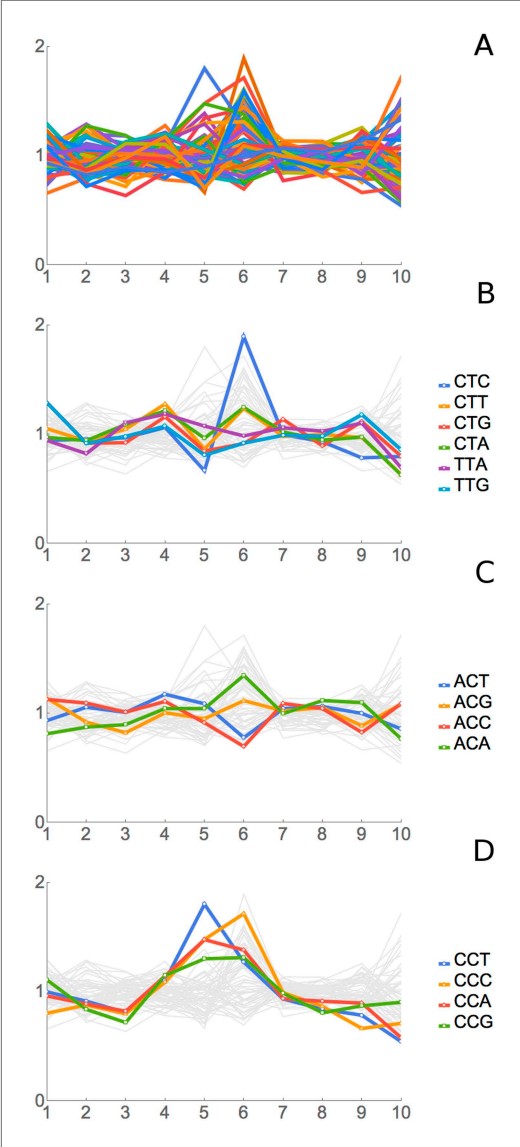

**Figure 4**. Results of Ribosome Residence Time analysis. (**A**) The pattern of RRTs for all codons at all positions. Most peaks are at position 6, with some at position 5. (**B**) The RRTs for the six leucine codons. CTC has the highest RRT of any codon at position 6. (**C**) The RRTs for the four threonine codons. ACC has the lowest RRT of any codon at position 6. (**D**) The RRTs for the four proline codons. Proline has peaks at position 5, the P-site, as well as at position 6.

To establish repeatability, we generated and analyzed three other ribosome profiling datasets and also re-analyzed previously published data (*Ingolia et al., 2009*). All five data sets gave qualitatively similar results; pairwise correlations for RRTs at position 6 ranged from 0.22 to 0.96 between the datasets (*Table 3*). The poorest correlation (0.22) was a correlation with the previously published dataset, which was generated using significantly different methods than our datasets. In particular, that dataset was generated by adding cycloheximide to the growing culture, then harvesting (*Ingolia et al., 2009*), whereas our data were generated by flash-freezing first, then adding cycloheximide to the frozen cells. Complete results for all five experiments are given in *Supplementary file 1*. More recently, we also subjected the long footprint data of *Lareau et al. (2014)* to RRT analysis and obtained correlations at position 6 of 0.21, 0.47, 0.23, and 0.27, respectively, for their 'untreated 1', 'untreated 2', 'untreated merge', and 'cycloheximide 1' experiments to our SC-lys experiment. Again, these experiments were carried out in a significantly different way from ours and it is not surprising that the correlations are modest. It is reassuring that a positive correlation can be seen even for experiments where no cycloheximide was used.

There are strong correlations between codon usage, the number of tRNA genes for the relevant tRNA, and tRNA abundance (*Ikemura, 1981*, *1982*; *Dong et al., 1996*; *Tuller et al., 2010*; *Novoa and Ribas de Pouplana, 2012*). Although one cannot determine causation from this correlation (*Plotkin and Kudla, 2011*), nevertheless it is consistent with the idea that the rate of decoding in translation is at least partly limited by tRNA concentration. Most of our results are consistent with this. However, there are some interesting exceptions. In yeast, the 61 sense codons are decoded by only 42 tRNAs. There are 12 pairs of codons that share a single tRNA (e.g., Phe TTC and TTT; Tyr TAT and TAC; etc) (*Roth, 2012*). In many but not all cases, the RRT of the two codons is similar (*Table 2*), consistent with the 'concentration' hypothesis. However, there are also cases where the RRT appears to be significantly different for two codons sharing the same tRNA. For instance, the Cys codon TGC has an RRT of 1.23, while TGT has an RRT of 0.81 (*Table 2*). Both codons are recognized by the same tRNA, which in this case is complementary for TGC, and wobble for TGT. Similarly, the Gly codon GGC has an RRT of 1.22 (tRNA is complementary), while GGT has an RRT of 0.93 (tRNA is wobble). Both these relationships (RRT for TGC > TGT, and RRT for GGC > GGT) were true in all five datasets (*Supplementary file 1*). In both the cases, the perfect match is decoded more slowly than the wobble match and in both cases, the slower, complementary pairing has a G:C match at the third (i.e., wobble) position. These and other similar examples (not shown) suggest that the RRT depends on more than just the concentration of the relevant tRNA. Perhaps the long RRTs for these

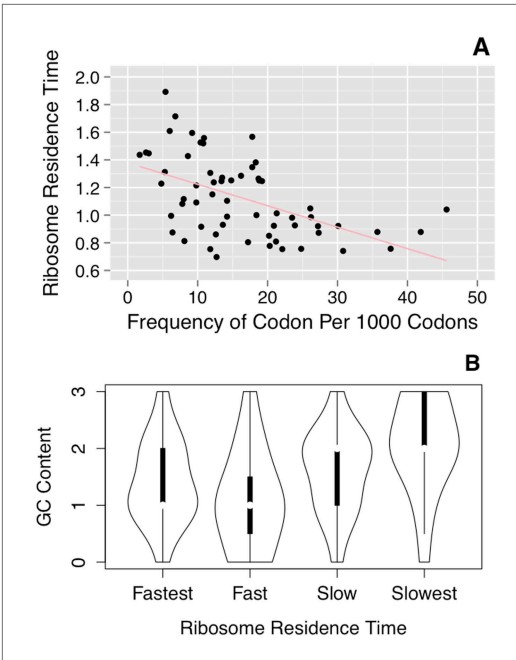

**Figure 5**. Correlation of ribosome residence times with codon properties. (**A**) Correlation of RRT with codon usage. RRT is plotted against the frequency of each codon per 1000 codons. (**B**) Correlation of RRT with the GC content of each codon. The codons were divided into quartiles by RRT (Fastest–Slowest), and the GC content of those ~15 codons is shown in a violin plot.

GC-rich codons are related to the time needed to eject incorrectly paired anti-codons of incorrect tRNAs, although this explanation is somewhat at odds with the literature (*Daviter et al., 2006*; *Gromadski et al., 2006*). Alternatively, it has been suggested that translocation can occur more quickly when the codon:anticodon interaction is weaker (*Semenkov et al., 2000*; *Khade and Joseph, 2011*).

## RRT analysis of short footprints
Recently, Lareau et al. made the exciting discovery that ribosome profiling on cells that have not been treated with any drug yields two classes of footprints, long (28–30 nucleotides) and short (20–22 nucleotides) (*Lareau et al., 2014*). It is the long class that is seen in cycloheximide experiments, and which we have characterized above. The short (20–22 nuc.) footprints seem to represent a different conformation of the ribosome, perhaps one that occurs when the ribosome translocates along the mRNA. Furthermore, Lareau et al. found that treatment of cells with the elongation inhibitor anisomycin efficiently generates short footprints. Lareau et al. suggest that the long and short footprints are reporting on two different states of translation (*Lareau et al., 2014*).

We applied RRT analysis to the short footprints generated by Lareau et al., with special focus on the footprints after anisomycin treatment. All three of their anisomycin datasets were studied, and the pairwise correlations between the RRT results for these three datasets were very high, ranging from 0.89 to 0.998. Partial results are shown in *Figure 7* and *Table 4*, and complete results are shown in *Supplementary file 2*. RRT analysis showed a series of peaks at different positions along the 7-codon footprint. The RRT values for the short footprints did not significantly correlate with RRT values for the long footprints, even when the phases of the footprints were shifted. This suggests, in agreement with Lareau et al., that the short and long footprints are indeed reporting on different translational processes. Furthermore, for the short footprints the RRT values are amino acid-specific, while for the long footprints at position 6, the RRT values are codon-specific (*Table 2*; *Table 4*; *Figure 4*, *Figure 7*, *Figure 8*). This again indicates that the two kinds of footprints are reporting on different translational processes. The amino acids in the peaks at positions 3, 5, and 6 are shown in *Table 4*: the peak at position 3 contains glycine; the peak at position 5 contains smallish hydrophobic amino acids (Leu, Val, Ile, and to some extent Phe), and the peak at position 6 is dominated by the two basic amino acids, Arg and Lys. It has previously been shown that basic amino acids can cause a pause in elongation by interacting with the ribosome exit tunnel (*Lu et al., 2007*; *Lu and Deutsch, 2008*; *Brandman et al., 2012*; *Wu et al., 2012*; *Charneski and Hurst, 2013*). The basis of the anisomycin arrest is partly but not fully understood (*Hansen et al., 2003*; *Blaha et al., 2008*), and so it is difficult to clearly interpret these results (but see 'Discussion'). Nevertheless, the application of RRT analysis to the anisomycin-generated footprints gives strong specific signals that are unlikely to be explained by a random process. We note, however, that results from the short footprints from untreated (no anisomycin) cells are only modestly correlated (0.23) with results from short footprints from the anisomycin-treated cells (data not shown).

It appeared that the RRT values at position 6 for the long footprints were codon-specific (*Figure 4*, *Table 2*), while the RRT values for the short footprints were amino acid-specific (*Figure 7*, *Table 4*). To confirm this, we developed a statistical test for the coherence of the results for a particular amino acid ('Materials and methods'). Briefly, this method tests whether every codon for a particular amino

**Table 2.** Ribosome residence time at position 6 (A) and 5 (B)

**A**

| Codon | AA | Usage | RRT | p value |
|---|---|---|---|---|
| CTC | Leu | 5.4 | 1.89 | *0.0001 |
| CCC | Pro | 6.8 | 1.71 | *0.0001 |
| GGG | Gly | 6 | 1.61 | *0.0001 |
| AGG | Arg | 9.2 | 1.59 | *0.0001 |
| ATA | Ile | 17.8 | 1.57 | *0.0001 |
| GGA | Gly | 10.9 | 1.56 | *0.0001 |
| TGG | Trp | 10.4 | 1.53 | *0.0001 |
| GTG | Val | 10.8 | 1.52 | *0.0001 |
| CGC | Arg | 2.6 | 1.45 | *0.0001 |
| CGA | Arg | 3 | 1.45 | *0.0008 |
| CGG | Arg | 1.7 | 1.44 | *0.0010 |
| TCG | Ser | 8.6 | 1.43 | *0.0001 |
| CCA | Pro | 18.3 | 1.38 | *0.0001 |
| ACA | Thr | 17.8 | 1.35 | *0.0001 |
| CCG | Pro | 5.3 | 1.31 | *0.0001 |
| GTA | Val | 11.8 | 1.31 | *0.0001 |
| GCA | Ala | 16.2 | 1.28 | *0.0001 |
| CCT | Pro | 13.5 | 1.27 | *0.0001 |
| TCA | Ser | 18.7 | 1.26 | *0.0001 |
| TAC | Tyr | 14.8 | 1.25 | *0.0001 |
| TAT | Tyr | 18.8 | 1.25 | *0.0001 |
| GAG | Glu | 19.2 | 1.25 | *0.0001 |
| CTA | Leu | 13.4 | 1.25 | *0.0001 |
| CTT | Leu | 12.3 | 1.24 | *0.0001 |
| TGC | Cys | 4.8 | 1.23 | *0.0001 |
| GGC | Gly | 9.8 | 1.22 | *0.0001 |
| CAG | Gln | 12.1 | 1.15 | *0.0002 |
| ACG | Thr | 8 | 1.12 | 0.0069 |
| AGT | Ser | 14.2 | 1.10 | 0.0060 |
| AGC | Ser | 9.8 | 1.09 | 0.0213 |
| CAC | His | 7.8 | 1.08 | 0.0098 |
| TTT | Phe | 26.1 | 1.05 | 0.0529 |
| GAA | Glu | 45.6 | 1.04 | 0.0538 |
| AGA | Arg | 21.3 | 1.01 | 0.3014 |
| TTC | Phe | 18.4 | 1.00 | 0.4955 |
| GCG | Ala | 6.2 | 0.99 | 0.4650 |
| TCC | Ser | 14.2 | 0.99 | 0.3341 |
| TTA | Leu | 26.2 | 0.99 | 0.3166 |
| TCC | Ser | 23.5 | 0.98 | 0.2249 |
| CAT | His | 13.6 | 0.93 | 0.0188 |
| GGT | Gly | 23.9 | 0.93 | *0.0003 |
| ATG | Met | 20.9 | 0.92 | 0.0027 |
| ATT | Ile | 30.1 | 0.92 | *0.0005 |

*Table 2. Continued on next page*

*Table 2. Continued*

**A**

| Codon | AA | Usage | RRT | p value |
|---|---|---|---|---|
| TTG | Leu | 27.2 | 0.92 | *0.0001 |
| CTG | Leu | 10.5 | 0.92 | 0.0139 |
| AAT | Asn | 35.7 | 0.88 | *0.0001 |
| AAA | Lys | 41.9 | 0.88 | *0.0003 |
| CGT | Arg | 6.4 | 0.87 | *0.0002 |
| CAA | Gln | 27.3 | 0.87 | *0.0001 |
| GCC | Ala | 12.6 | 0.86 | *0.0001 |
| GAC | Asp | 20.2 | 0.85 | *0.0001 |
| TGT | Cys | 8.1 | 0.81 | *0.0001 |
| GCT | Ala | 21.2 | 0.81 | *0.0001 |
| ATC | Ile | 17.2 | 0.80 | *0.0001 |
| ACT | Thr | 20.3 | 0.78 | *0.0001 |
| GAT | Asp | 37.6 | 0.76 | *0.0001 |
| AAC | Asn | 24.8 | 0.76 | *0.0001 |
| GTT | Val | 22.1 | 0.75 | *0.0001 |
| GTC | Val | 11.8 | 0.75 | *0.0001 |
| AAG | Lys | 30.8 | 0.74 | *0.0001 |
| ACC | Thr | 12.7 | 0.70 | *0.0001 |

**B**

| Codon | AA | Usage | RRT | p value |
|---|---|---|---|---|
| CCT | Pro | 13.5 | 1.80 | *0.0001 |
| CCC | Pro | 6.8 | 1.48 | *0.0001 |
| CCA | Pro | 18.3 | 1.48 | *0.0001 |
| AAT | Asn | 35.7 | 1.39 | *0.0001 |
| CGC | Arg | 1.7 | 1.34 | 0.0070 |
| CCG | Pro | 5.3 | 1.30 | *0.0001 |

A. Usage of each codon per 1000 codons and the Ribosome Residence Time (RRT) at position 6 (the A-site of the ribosome). The p-value for a difference between the calculated RRT value and an RRT value of 1 is shown. p-values less than or equal to 0.001 are marked with an asterisk. B. As for A, but for the six highest values at position 5 (the P-site).

acid behaves similarly, and it yields a small p-value if it does. Indeed, this analysis confirms that the short footprints give results specific to the amino acid, while the long footprints generally do not (i.e., the long footprints are codon-specific) (*Figure 8*). This suggests that the long footprints are reporting on the process of decoding (which depends on specific codons), while the short footprints are reporting on events after decoding.

## Discussion

To our knowledge, this is the first measurement of the differential rate of translation of all 61 codons in vivo. There is a correlation between a high codon usage and a high rate of decoding. Although this is a correlation that has been widely expected, there has been little evidence for it; indeed, the most recent experiments suggested that all codons were decoded at the same rate (*Qian et al., 2012*; *Charneski and Hurst, 2013*). Some workers have had other expectations for decoding rates. For instance, an important theory was that the more common codons were common because their translation might be more accurate (*Plotkin and Kudla, 2011*) (and this still might be correct).

Translation is optimized for both speed and accuracy (*Bieling et al., 2006*). During translation, the ribosome must sample many incorrect tRNAs at the A-site before finding a correct tRNA. It must

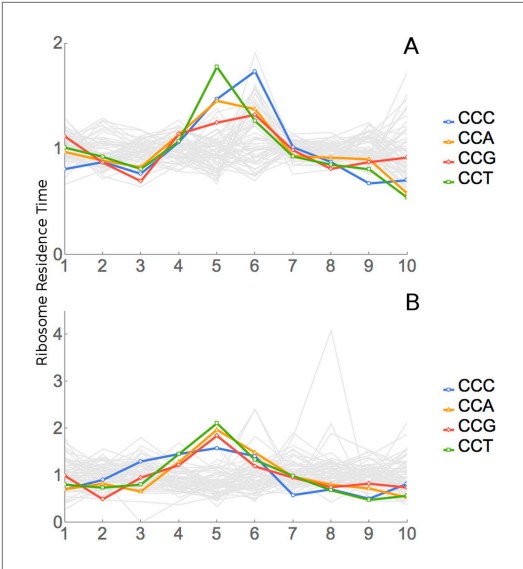

**Figure 6**. Analysis of ProPro dipeptides. (**A**) RRT analysis of windows containing no ProPro dipeptides. (**B**) RRT analysis of windows containing ProPro dipeptides.

match the anti-codon of that correct tRNA with the codon; after such matching, there is a conformational change around the codon–anticodon interaction at the decoding center (*Demeshkina et al., 2012*; *Zeng et al., 2014*). The ribosome must form the peptide bond (*Rodnina, 2013*; *Polikanov et al., 2014*), translocate (*Semenkov et al., 2000*; *Khade and Joseph, 2011*; *Zhou et al., 2014*), and eject the empty tRNA. The nascent peptide must make its way through the ribosome exit tunnel (*Lu and Deutsch, 2008*; *Petrone et al., 2008*; *Lu et al., 2011*; *Wilson and Beckmann, 2011*). Depending on the rate of each of these events, the concentration of the various tRNAs might or might not have a detectable effect on the overall rate of translation. Our findings that (i) the more frequent codons (i.e., the ones with the highest tRNA concentrations) are decoded rapidly; and (ii) GC-rich codons are decoded slowly; and (iii) proline is slow in the P-site, suggest that there are at least three processes that happen somewhat slowly and on a similar timescale. The high rate of decoding for high concentration tRNAs may reflect the relatively short time

it takes for the ribosome to find a high-concentration correct tRNA among many incorrect tRNAs. The fact that we detect proline-specific delays of a similar magnitude to the rare-codon specific delays suggest that peptide bond formation and identification of the correct tRNA are happening on similar time scales. In general, this is what one might expect from the evolution of such an important process as protein synthesis—if one process was entirely rate-limiting, there would be very strong selection for greater speed in that process, until a point is reached where it 'catches up' with other processes, and several processes together are then rate-limiting.

Even though these data establish that common codons are translated relatively rapidly, this does not on its own explain the success of codon optimization for increasing protein expression, since the rate of translation is primarily limited by the rate of initiation, not elongation (*Andersson and Kurland, 1990*; *Plotkin and Kudla, 2011*) (although one recent study identifies a mechanism whereby rapid elongation causes rapid initiation [*Chu et al., 2014*]). Nevertheless, on a genome-wide (and not gene-specific) scale, the use of faster codons would mean that a given genomic set of mRNAs would require (or titrate out) fewer ribosomes to make a given amount of protein than the same set of mRNAs using slower codons (*Andersson and Kurland, 1990*; *Plotkin and Kudla, 2011*). Based on our RRT

**Table 3.** Correlations between experiments

|  | YPD1 | -His | YPD2 | Ingo. |
|---|---|---|---|---|
| -Lys | 0.80 | 0.35 | 0.76 | 0.22 |
| YPD1 |  | 0.53 | 0.96 | 0.55 |
| -His |  |  | 0.58 | 0.37 |
| YPD2 |  |  |  | 0.53 |

The pairwise Spearman correlations between the RRT values at position 6 are shown for five independent experiments, where the experiments are named YPD1, YPD2, SC-Lys, SC-His, and Ingolia. The SC-Lys and SC-His experiments were carried out by JG, and used flash-freezing as the initial method for stopping ribosome movement. The YPD1 and YPD2 experiments were carried out by YC (*Cai and Futcher, 2013*), and used addition of ice and cycloheximide to the culture as the initial method for stopping ribosome movement. The 'Ingo' experiment was that carried out by *Ingolia et al. (2009)*. Further details are given in 'Materials and methods'. Complete RRT values for each position in each experiment are provided in *Supplementary file 1*.

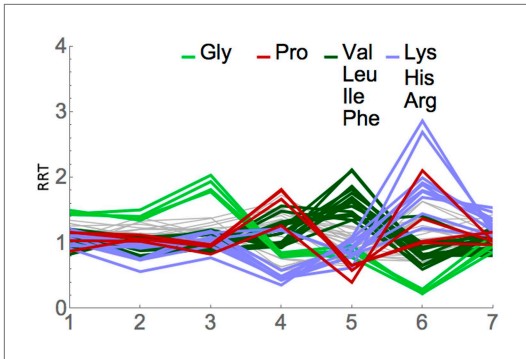

**Figure 7**. RRT analysis of short footprints from anisomycin treatment. The short, seven-codon footprints from anisomycin treatment (dataset 1b) from *Lareau et al. (2014)* were analyzed for RRT. All 61 sense codons are shown; codons for selected amino acids are color-coded by amino acid. Position along the footprint is shown on the x-axis.

measurements, and taking into account the different copy numbers of different mRNAs (*Lipson et al., 2009*), we roughly estimate that yeast requires about 5% fewer ribosomes than if they were to make protein at the same overall rate but using each synonymous codon at an equal frequency ('Materials and methods'). This provides at least a sufficient reason for the bias towards faster synonymous codons.

We applied RRT analysis to the short footprints identified by Lareau et al. (*Figure 7*). These short footprints seem to report on a different translational process than the long footprints seen in cycloheximide experiments. We see that the basic amino acids Arg and Lys are slow at position 6; small hydrophobic amino acids are slow at position 5; and glycine is slow at position 3. While we know too little about the nature of the short footprints to reliably interpret these results, one speculative possibility is that the results report on the interaction of amino acids in the nascent peptide chain with the exit tunnel of the ribosome (*Raue et al., 2007*; *Petrone et al., 2008*; *Berndt et al., 2009*; *Bhushan et al., 2010*; *Lu et al., 2011*; *Wilson and Beckmann, 2011*; *Gumbart et al., 2012*). We find Arg and Lys slow at position 6, and this correlates with the fact that these basic amino acids cause a pause by interacting with the exit tunnel (*Lu et al., 2007*; *Lu and Deutsch, 2008*; *Brandman et al., 2012*; *Wu et al., 2012*; *Charneski and Hurst, 2013*). This would then suggest that small hydrophobic amino acids, and then glycine, might similarly cause pauses by interacting with positions one or three amino acids further out in the exit tunnel.

In summary, we believe that RRT analysis is a sensitive high-resolution method that can characterize the interaction of codons and amino acids with the ribosome. It can be applied to ribosome profiling data of many types, from many organisms. In this study, we show that frequent codons are decoded more quickly than rare codons; that codons high in AT are decoded somewhat quickly; that proline forms peptide bonds slowly; and that short footprints from anisomycin treated cells have an interesting RRT profile that may reflect interaction of amino acids with the ribosome exit tunnel.

## Materials and methods

Experiments were done with yeast strain background BY4741. Ribosome profiling was based on the method of Ingolia (*Ingolia et al., 2009*), but with modifications (see below). Programs for analysis of

**Table 4.** Top 10 RRTs at positions 3 through 6 of the anisomycin-generated short footprints

| Pos 3 | Pos 4 | Pos 5 | Pos 6 |
|---|---|---|---|
| Gly GGG 2.64 | Pro CCC 2.36 | Leu TTA 2.75 | Arg CGA 3.72 |
| Gly GGC 2.52 | Pro CCA 2.34 | Leu CTC 2.73 | Arg CGG 3.50 |
| Gly GGT 2.36 | Met ATG 2.25 | Val GTA 2.43 | Pro CCG 2.74 |
| Gly GGA 2.32 | Pro CCT 2.17 | Leu CTA 2.36 | Lys AAA 2.59 |
| Asp GAC 1.80 | Ala GCC 2.13 | Leu TTG 2.29 | Lys AAG 2.49 |
| Ala GCC 1.79 | Phe TTC 2.03 | Val GTG 2.21 | Arg CGC 2.46 |
| Ala GCA 1.70 | Ala GCA 2.01 | Leu CTT 2.16 | Arg CGT 2.34 |
| Ala GCT 1.65 | Ala GCT 1.98 | Val GTC 2.12 | Arg AGG 2.32 |
| Ala GCG 1.59 | Tyr TAC 1.98 | Val GTT 2.11 | Arg AGA 2.21 |
| Blu GAG 1.58 | Ser TCC 1.97 | Ile ATA 2.03 | Asp GAT 2.12 |

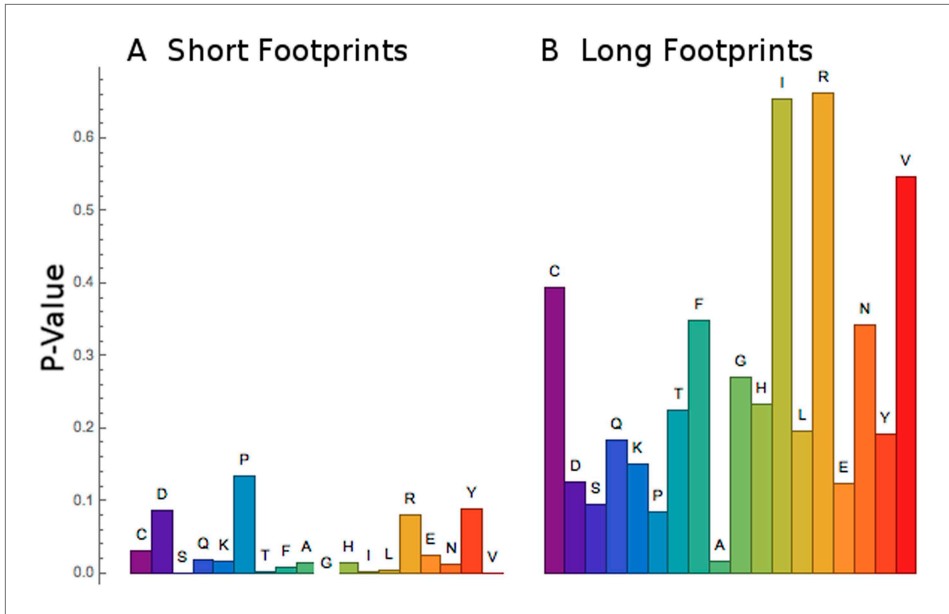

**Figure 8**. Short footprints are amino acid-specific; long footprints are codon-specific. For the set of codons corresponding to each amino acid (x-axis), a test was done to see if all the codons behaved similarly or not. For the short footprints (left, panel **A**), p-values (y-axis) are generally small, showing that each codon for a particular amino acid behaves similarly ('Materials and methods'). For the long footprints (right, panel **B**), p-values are generally large, showing that the codons for each particular amino acid behave differently ('Materials and methods').

ribosome residence time were written by the authors, primarily RY and AY. The Perl code for ribosome residence time analysis is given in *Source code 1 and 2*.

## Ribosome profiling

Informatic analysis was conducted on four ribosome profiling experiments (YPD1, YPD2, SC-lys, and SC-his) done for other reasons in the Futcher lab. The strains and methods used varied slightly from experiment to experiment; nevertheless similar results were obtained for the RRT analysis (*Table 2*). The ribosome profiling experiments YPD1 and YPD2 have been reported previously (*Cai and Futcher, 2013*) as the 'WT' and 'whi3' experiments, respectively.

All experiments used *S. cerevisiae* strain background BY4741. Two biologically independent ribosome-profiling libraries and mRNA-seq libraries were obtained from YPD rich media (the YPD1 and YPD2 experiments), and two biologically independent ribosome-profiling libraries and mRNA-seq libraries were prepared in synthetic media (the SC-lys and SC-his experiments). Two methods for harvesting cells were used. After harvesting and footprint size selection, footprints from all four experiments were processed identically into sequencing libraries using the ARTseq Yeast Ribosome Profiling kit, following the manufacture's instructions beginning with step B3 in the protocol.

### Harvesting method 1 (YPD1 and YPD2 experiments)

1 liter of cells in YPD were grown to a density of $2.0 \times 10^7$ cells/ml. Medium was cooled to 0°C by adding ice (stored at −20°C) and simultaneously cycloheximide was added to a concentration of 100 μg/ml to quickly halt translation and freeze translating ribosomes in place. Cells were centrifuged using a Sorvall Evolution RC centrifuge at 3000 rpm for 2 min at 4°C. The resulting cell pellet was washed with ice-cold RNase-free water containing 100 μg/ml cycloheximide by gentle vortexing and repelleted. Supernatant was aspirated, and cells were resuspended in polysome lysis buffer prepared according to the ARTseq ribosome profiling kit instructions. Cell lysis buffer slurry was slowly dripped into an RNase-free 50 ml conical tube containing liquid nitrogen. Resulting frozen pellets of cell slurry were lysed using a TissueLyser II and 50 ml grinding jars at liquid nitrogen temperature for six 3 min cycles

at 15 hertz. Frozen cell lysate was scraped from the grinding jar into a new RNase-free 50 ml conical tube followed by reheating the slurry in a 30°C water bath with constant swirling. Immediately after complete thawing (~3–5 min), cell lysate was centrifuged for 5 min at 3000×$g$. Supernatant was moved to a 1.5 ml RNase-free centrifuge tube and centrifuged for 10 min at 20,000×$g$. Clarified lysate total RNA content was estimated using a Nanodrop at A260 nm, and polysome complexes were digested using ARTseq ribonuclease mix according to the manufacture's instructions. Ribosome-protected mRNA footprints were purified using an Illustra Microspin S-400HR column prepared according to ARTseq manufacture's instructions. All following library generation steps were performed according to the ARTseq protocol starting at step 4 (PAGE purification). Following the end repair step in the protocol, a biotinylated oligonucleotide antisense to a specific rRNA fragment was used to reduce rRNA contamination using a protocol from the Jonathan Weissman lab (personal communication from Gloria Brar).

## Harvesting method 2 (SC-lys and SC-his experiments)

Synthetic media lacking lysine or lacking histidine was used to prepare 1 liter of cells at $2.0 \times 10^7$ cells/ml. The strains were prototropic for Lys or His (*HIS3* gap1 frame1), respectively. Cells were harvested by vacuum filtration using Whatman 7184–009 membrane filters at 30°C. A liquid nitrogen cooled spatula was used to scrap cells from the membrane followed by immediate flash freezing in an RNase-free 50 ml conical tube containing liquid nitrogen. Special care was taken to ensure cells were exposed to air for as little time as possible, between vacuum filtration and flash freezing (2–3 s), to prevent the loss of ribosome footprints at the 5′ ends of mRNAs (personal communication, Gloria Brar). ARTseq polysome lysis buffer containing cycloheximide at 50 µg/ml was slowly dripped into the liquid nitrogen filled cell pellet conical tube. Cells were lysed using a TissueLyser II and 50 ml grinding jars at liquid nitrogen temperature for six 3 min cycles at 15 hertz. Frozen cell lysate was scraped from the grinding jar into a new RNase-free 50 ml conical tube followed by reheating the slurry in a 30°C water bath with constant swirling. Immediately after complete thawing (~3–5 min), cell lysate was centrifuged for 5 min at 3000×$g$. Supernatant was moved to a 1.5 ml RNase-free centrifuge tube and centrifuged for 10 min at 20,000×$g$. Clarified lysate total RNA content was estimated using a Nanodrop at A260 nm, and polysome complexes were digested using ARTseq ribonuclease mix according to the manufacture's instructions.

## SC-lys Dataset

Digested monosomes were purified using sucrose cushion ultracentrifugation for 3 hr at 35,000 rpm using a SW-41 rotor. The sucrose cushion contained 9 ml of 10% sucrose polysome lysis buffer lacking triton detergent layered over 3 ml of 60% sucrose polysome lysis buffer lacking triton detergent. Gradient fractionation was carried out using a BioRad EM-1 UV absorbance monitor and a peristaltic pump. Efficiency of RNase digestion was monitored in tandem using an undigested control lysate on an identically prepared 10–60% sucrose cushion and a digested control centrifuged on a 10–60% sucrose gradient. Following fractionation, the monosome containing fraction was mixed 1:1 with 4 M guanidine thiocyanate and was precipitated overnight using a 1:1 vol of 100% isopropanol chilled to −20°C. The RNA pellet was aspirated and resuspended in 400 µl RNase-free water, and protein was removed by two acid phenol–chloroform purifications followed by one chloroform purification. Recovered supernatant was brought to 0.3 M ammonium acetate and precipitated with 3 vol of 100% ethanol. All following library generation steps were performed according to the ARTseq protocol starting at step 4 (PAGE purification). Following the end repair step in the protocol, a biotinylated oligonucleotide antisense to a specific rRNA fragment was used to reduce rRNA contamination using a protocol from the Jonathan Weissman lab (personal communication Gloria Brar).

## SC-his Dataset

Digested monosomes were purified using an Illustra Microspin S-400HR column according to ARTseq manufacture's instruction. All following library generation steps were performed according to the ARTseq protocol starting at step 4 (PAGE purification). Following the end repair step in the protocol, a biotinylated oligonucleotide antisense to a specific rRNA fragment was used to reduce rRNA contamination using a protocol from the Jonathan Weissman lab (personal communication Gloria Brar).

## Data analysis

Unless indicated, data processing and analysis were performed using a collection of custom programs written in Perl.

## Sequence processing and alignment

Primary data were generated using Illumina HiSeq2000. Data were processed using Fastq clipper from the FASTX Toolkit 0.0.13 to remove the adaptor sequence and all reads shorter than 25 nucleotides were discarded. Alignment to the reference was done using bowtie2 2.1.0 in local alignment mode.

Before performing our analysis on the *Ingolia et al. (2009)* data, in order to adhere to the processing guidelines of that paper, we used bowtie 0.12.8, reporting all alignments with at most three mismatches, and a seed length of 21. We then processed the multiple alignments, removing the poly-A tails and picking the one with the greatest number of bases matching to the reference.

## Ribosome residence time analysis

This analysis uses the general idea that many different mRNA sequences should get an independent and equal vote on decoding speed. We opted to analyze select regions where the effects of codon usage become particularly easy to assay. First, we discounted all reads with more than two mismatches or quality less than 10. We identified the first in-frame codon of each read and discarded those less than 30 nucleotides long to exclude fragments that may have been over digested by RNAase I. We then examined the coding regions of the genome, ignoring those overlapping with other genes, rRNAs, and tRNAs, in order to maximize our confidence in unique mapping. Each of the footprint reads that fully fit into a coding region that it aligned to was considered for further analysis.

For each particular codon, we identified all instances in our coding regions where this codon (say CTC) occurs uniquely within a window of 10 codons upstream and 10 codons downstream (i.e., a window of 19 codons with the target CTC in the center of the window). For footprints that are 10 codons long, there will be 10 classes of footprints where this particular CTC can appear—position 1, position 2, ..., position 10. Thus, all footprints where the first codon of the footprint aligns to this particular CTC will belong to the position 1 class, all footprints where the second codon of the footprint aligns to this particular CTC will belong to position 2 class, etc.

In the absence of any codon preference of the ribosome, we would expect to see a uniform distribution of reads across these 10 classes. In general, the codon-positional preference is described by the relative frequency of reads in each of these classes. These relative frequency distributions can be fairly averaged over all target regions over all genes centered on a specific codon. This average we call the 'Ribosome Residence Time' (RRT); it is intended as a statistical estimate of the relative time spent by the ribosome at a particular codon at a particular position. Typically we discuss the RRT at position 6 (the A-site), but we also discuss the RRT at position 5 (the P-site). Regions on highly expressed genes can be fairly compared with similar regions on genes with lower expression, because we are dealing with relative frequency distributions (i.e., percentage instead of read counts). Each region represents an independent trial of any positional preference of the given central codon. Averaging over the 100s or 1000s of occurrences on the genome provides for a statistically rigorous analysis.

Relative frequency distributions will only be representative if the observed number of reads in the window is high enough that no single position dominates the distribution. For this reason, we restricted our analysis to windows with at least 20 total reads with at least 3 non-empty classes.

The frequency distributions are not normally distributed; this is in part because the number of reads is limited, so many windows have zero footprints at many positions, so the mode of the distribution is often 0. Nevertheless we believe that the mean is a good summary statistic. Maximum values are less than 1, so the mean cannot be skewed by extremely high values. We have also calculated the RRTs using the median of the windows instead of the mean, but the results are almost indistinguishable. The Spearman rank correlation between the RRTs as calculated by the mean, and by the median, is 0.97, while the Kendall Tau correlation is 0.89.

For each codon, we obtain the two-tailed p-value by comparing the experimentally determined relative frequency to the distribution of 10,000 relative frequencies based on permuted results. For

each of the 10,000 instances, for each considered window, we permute the footprint counts of the 10 position classes.

We performed our RRT analysis on the *Ingolia et al. (2009)* data, with small modifications. We did not perform the checks of read quality and the number of mismatches, as this was taken care of in pre-processing steps (See Sequence Processing and Alignment). We also considered all reads with at least 24 nucleotides and performed our relative frequency calculations on the eight codons, because the majority of the reads were shorter than the reported size selection of RNA fragments ~27–31 nucleotides in length.

The statistical significances shown in *Table 1* were obtained by constructing 10,000 simulated frequency distributions by randomly and independently permuting each region's frequency distribution prior to averaging. The rank of each observed positional peak among these simulated distributions established the p-value.

## Codon coherence analysis

We developed a p-value computation to assess whether the codons for a given amino acid behave similar to one another (i.e., are coherent) or not. Each codon's RRT values along the positions of a footprint may be considered as a k-dimensional vector, where k is the number of positions in the footprint (10 for long reads vs 7 for short reads). We consider the position in k-dimensional space of the end-point of this vector. For the set of synonymous codons for a particular amino acid, we consider the set of endpoints. For any given set of c such endpoints, we can compute the average pairwise distance d between them over all $c(c-1)/2$ pairs of points. If all codons for an amino acid behave similarly, then the endpoints are close together, and the distance d is relatively small, indicating codon coherence (amino-acid specific behavior), whereas if the various codons for a given amino acid behave differently (non-coherence, codon-specific behavior), then the distance d is relatively large.

To judge the sizes of these distances for a particular set of points, S, containing c codons (c ranges from 2 to 6) for a particular amino acid, we use a p-value. We construct 10,000 random samples of c codons drawn from the 61 possible sense codons. For each sample, we compute the average pairwise distance and compare this to the average pair distance of S. The rank of S in this distribution provides a p-value, which is significant if the vast bulk of random samples have greater pairwise distance than S. Results are shown in *Figure 8*.

## Estimates of ribosomes needed for differently-encoded transcriptomes

An mRNA encoding a given protein could use only the fastest codon for each amino acid or only the slowest or it could use a mixture. In each case, the mRNA would occupy, or titrate out, a different number of ribosomes. A transcriptome of mRNAs using only the slowest codons would require more ribosomes to make a given amount of total protein in a given time than a transcriptome of mRNAs using only the fastest codons. We roughly estimated the size of this effect for the range of codon decoding speeds we observed. We generated *in silico* a yeast transcriptome using only the fastest codon for each amino acid at position 6 (from *Table 1*) or only the slowest codon or a random mixture of codons. Furthermore, we weighted the abundance of each mRNA according to its actual abundance as measured by *Lipson et al. (2009)*. We then compared the relative time required to translate each of these *in silico* transcriptomes by a set number of ribosomes based on the RRT values for each codon at position 5 and 6, and also assuming that the relevant delay is the delay at position 5 plus the delay at position 6 (since these two reactions must occur sequentially and not simultaneously before the ribosome can shift along the mRNA). In doing this, we noted that the RRT values for position 5 are negatively correlated with those at position 6. Results are as follows: the random encoding requires 1.050 as long as WT; the slowest encoding requires 1.168 as long as WT; and the fastest encoding requires 0.930 as long as WT. Note that this estimate uses the simplification that each species of mRNA will initiate translation at the same rate. A more accurate calculation in which the more abundant mRNAs initiate more rapidly than average would increase the difference between the WT and the random encodings.

## Note added in proof

When the accepted manuscript was published, RRT values from an earlier version of the algorithm were erroneously used for *Figure 5* (but not for other figures), giving a correlation of –0.7 between RRT and codon usage. The current algorithm, used here, gives a corrected version of *Figure 5*, shown here, with a correlation of –0.52.

## Acknowledgements

We thank J Weissman and G Brar for their generosity in helping us learn ribosome profiling and for providing protocols and advice. Three anonymous reviewers provided insightful comments that greatly improved the final manuscript. This work was supported by NIH grant R01 GM098400 to BF and NSF grants DBI-1060572 and IIS-1017181 to SS.

## Additional information

### Funding

| Funder | Grant reference number | Author |
| --- | --- | --- |
| National Institute of General Medical Sciences | RO1 GM098400 | Bruce Futcher |
| Directorate for Computer and Information Science and Engineering | DBI-1060572 | Steve Skiena |
| Directorate for Computer and Information Science and Engineering | IIS-1017181 | Steve Skiena |

The funders had no role in study design, data collection and interpretation, or the decision to submit the work for publication.

### Author contributions

JG, Conception and design, Acquisition of data, Analysis and interpretation of data, Drafting or revising the article; RY, Wrote code., Conception and design, Analysis and interpretation of data; AY, Wrote code., Conception and design, Analysis and interpretation of data, Drafting or revising the article; YC, Acquisition of data; SS, Designed algorithm., Conception and design, Analysis and interpretation of data, Drafting or revising the article; BF, Conception and design, Analysis and interpretation of data, Drafting or revising the article

### Author ORCIDs

Bruce Futcher, http://orcid.org/0000-0002-1012-9022

## Additional files

### Supplementary files

• Supplementary file 1. Complete Ribosome Residence Times for each codon at each of the 10 possible codon positions in a 30 nt (or, for Ingolia data, 24 nt) ribosome footprint. Each Excel spreadsheet is based on data from an independent biological experiment. Four of these experiments were done during the course of this work, two experiments by JG and two experiments by YC, while the fifth experiment was published by *Ingolia et al. (2009)*. (**A**) Ribosome Residence Time analysis for all codons from the SC-lys expt. (**B**) Ribosome Residence Time analysis from the YPD1(WT) expt. (**C**) Ribosome Residence Time analysis from the YPD2 (whi3) expt. (**D**) Ribosome Residence Time analysis from the SC-his expt. (**E**) Ribosome Residence Time analysis from the Ingolia expt.

• Supplementary file 2. Complete Ribosome Residence Times for each codon at each of the 7 possible codon positions in a 21 nt ribosome footprint. Each Table is based on one of the three anisomycin datasets of *Lareau et al. (2014)*. (**A**) RRT for short footprints; aniso2 dataset. (**B**) RRT for short footprints; aniso1B dataset. (**C**) RRT for short footprints; aniso1A dataset.

• Source code 1. Source code 1 is a plain text file containing stage 1 of the Perl code for Ribosome Residence Time analysis.

• Source code 2. Source code 2 is a plain text file containing stage 2 of the Perl code for Ribosome Residence Time analysis.

## Major datasets

The following dataset was generated:

| Author(s) | Year | Dataset title | Dataset ID and/or URL | Database, license, and accessibility information |
|---|---|---|---|---|
| Gardin, et al., | 2014 | Measurement of average decoding rates of the 61 sense codons in vivo | SRP044053 | NCBI SRA database. |

The following previously published dataset was used:

| Author(s) | Year | Dataset title | Dataset ID and/or URL | Database, license, and accessibility information |
|---|---|---|---|---|
| Ying Cai | 2013 | Ribosome profiling of whi3 mutant yeast | www.ncbi.nlm.nih.gov/geo/query/acc.cgi?acc=GSE51164 | Publicly available at NCBI Gene Expression Omnibus. |

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
