## [Decision Letter]

Thank you for sending your work entitled “Measurement of decoding rates of all
individual codons in vivo” for consideration at *eLife*. Your
article has been favorably evaluated by Aviv Regev (Senior editor), a Reviewing editor,
and 3 reviewers.

The Reviewing editor and the reviewers discussed their comments before we reached this
decision, and the Reviewing editor has assembled the following comments to help you
prepare a revised submission.

Your manuscript addresses the question of variation in average decoding time for all the
tRNAs in the yeast *Saccharomyces cerevisiae*, by using ribosome
footprinting at codon resolution. It describes a novel statistics (Ribosome Residence
Times, RRT) that was used for four ribosome profiling datasets obtained from *S.
cerevisiae*. You show that the RRTs correlate with codon usage and therefore
suggest that RRT could be used to characterize the decoding rates of codons that have
the same sequence.

Two of the reviewers, who are experts in protein synthesis, but not in
statistics/bioinformatics, wrote favorable reviews on your work. However, the third
reviewer who is more knowledgeable in statistics/bioinformatics was rather critical, as
detailed below. A major appeal of your work is that you arrive at a different conclusion
than previous work Qian et al and Charneski & Hurst, using ribosome-profiling data,
which has concluded that there is little or no difference in the rates of decoding by
tRNAs. The latter conclusions contradict a large body of previous work in genetics,
molecular biology and biochemistry of translation, which clearly showed that there are
significant differences in the rates of decoding. This naturally led to a significant
amount of confusion in the field. You suggest that the abundant transient pausing on
mRNAs, in the previous published study, caused by other effects, probably made it
impossible to see the relatively more subtle differences in ribosome residence time
derived from differences in codon recognition.

The major comments you need to address are as follows:

1) The authors suggest that AT rich codons are decoded more rapidly than GC rich codons
but this is not clearly shown in the manuscript. At first this seemed counterintuitive
but the authors suggest an interesting possibility that for GC rich codons incorrect
tRNAs might dwell in the A site longer than they would at weaker codons and the
aggregate of those non-productive interactions might increase the step time at these
codons. The authors do not cite any biochemical studies that would support this
conclusion and in fact they do not cite previous biochemical studies in the manuscript
in many places where that would be appropriate.

2) The manuscript at many places simply states conclusions without providing any
reference (where that would be appropriate) or argument for the conclusion.
Substantiation of the conclusions must be included.

3) The major critique of the more critical reviewer is as follows: “Two of the
datasets used in the manuscript were previously published by the same group and two have
not been published before, however the authors state that the other two were also
obtained ”for other reasons in the Futcher lab“. Therefore while the
manuscript provides some newly generated experimental data, the primary focus is on
computational analysis and in particular on RRT. How valuable and useful is RRT? This,
unfortunately, is not very clear. The very fact that it correlates with codon usage
merits further investigation as it may indeed provide potentially useful characteristic
for decoding rates distributions of the codons with the same sequence. Unfortunately RRT
is poorly characterized in the manuscript. Its relationship to the decoding is not
explored beyond correlations with codon usage. Also the manuscript misleadingly treats
RRT as a measure (rather than potentially related statistics) of codon decoding
rates.”

4) The title “Measurement of decoding rates of all individual codons in
vivo” is highly misleading.

Lys codon AAA at a position i of mRNA x is not the same as Lys codon AAA at a position j
of mRNA y. AAAix and AAAjy likely would have different decoding rates. It would be
wonderful to be able to measure decoding rates of individual codons, but this is not the
case here. RRT is a relative footprint density of individual codons within a window of
10 neighbor codons averaged over all codons that share the same sequence.

5) Before averaging, it makes sense to show that the average is a useful characteristic
of the distribution. It would be the case if the distributions were normal or at least
could be approximated to normal, as this might not be the case.

If the distributions are not normal, the authors may explore other descriptive
statistics of the distribution (for example median) for their relationship with codon
usage, tRNA gene copy number or tAI. But it is highly important to obtain descriptive
statistics of the distributions first. Even if the distributions are normal, the
dispersion of these distributions may not be the same.

6) The procedure for RRT may have a hidden relationship with the codon usage, thus
explaining the observed correlation. This could be the case because of intrinsic
non-randomness of codon sequences, which effect codon distributions within the 10-codon
window. Say a codon X may appear more frequently in the windows centered around the
codon Y than in the windows centered around the codon Z. To explore whether there is a
hidden relationship to codon usage, the authors should assign experimental footprint
densities to codons randomly. Then calculate RRTs and explore how obtained RRT values
relate to the codon usage.

In addition to that it would make sense to carry out RRT calculations for naked mRNA
controls and explore obtained distributions in a similar manner.

7) Does RRT really relate to codon decoding rates? Intuitively it should, but this needs
to be shown. We know from a number of experiments, including ribosome profiling that
concentration of tRNAs effect the decoding rates. There are publicly available datasets
where particular aminoacylated tRNAs were depleted, see for example [34] eLife [PMID: 24842990] for S.
cerevisiae or [35] Nature
[PMID:22456704] for bacterial organisms. There might be more. The authors should
calculate RRT for these datasets and characterize RRTs between these datasets.

8) If RRT is indeed a good statistics it should work well not only in yeast, but also in
other organisms, thus it is important to carry similar analysis on other publicly
available datasets. The authors did that for the very first ribosomal profiling dataset
[28] dataset, which was
generated with cycloheximide pretreatment. It also has a relatively poor coverage. For
yeast it would be advisable to use [34] Elife [PMID: 24842990] data.

RRT should also correlate with other ways of measuring elongation speed. For example,
the authors could calculate RRT for [29] Cell [PMID: 22056041] and explore whether an aggregated RRTs for the
codons of individual coding sequences could predict differences in the speed of
ribosomes over individual mRNAs estimated with the pulse chase experiment described in
this work.

9) The critical reviewer is skeptical regarding the use of ribosomal profiling data for
estimating decoding rates for several reasons. The time that ribosome spends at a
particular codon is only one of the factors affecting the number of footprints aligning
to that codon. The others are sequence coverage (i), initiation rate for the start codon
of corresponding ORF (ii) and concentration of corresponding mRNA (iii). These factors
can be estimated (e.g. mRNA-seq can estimate relative mRNA levels) or the data could be
normalized in the ways that would minimize the contribution of these factors, e.g. a
footprint density at a particular location can be normalized over the cumulative density
for the entire dataset or for individual mRNAs. Such normalization procedures
aren't perfect and may generate certain artifacts and Gardin et al do discuss it
rightfully to some extent. However, there are other factors that are more difficult to
take into account, such as the effect of antibiotics on capture of the ribosomes at
particular locations or even at specific ribosomal conformations leading to differing
length of footprints, see [34]
*eLife* [PMID: 24842990] for more information. The reviewer is also
surprised and disappointed that this very important and highly relevant article is not
mentioned in the manuscript.

The other factors effecting densities are those related to the biases of cDNA library
preparation. Their presence can be easily seen in the data analyzed here with RRT as
well, e.g. Figure 2; codon 10 corresponds to the
5' ends of the footprints and show a variability comparable to that of position 6
(btw plotting 64 curves on the same diagram is not very effective, the authors should
explore other statistics for measuring variability within a distribution). Most likely
this variability is due to sequence specificity of RNAse cleavage and/or adapter
ligation. The other factor that is highly relevant to measuring decoding rates is PCR
amplification of cDNA libraries. PCR amplifies fragments non-linearly, and the ratio
between a low abounded fragment and a highly abounded fragment would likely increase
after PCR. To control for that the initial step of cDNA amplification should be carried
out with RT primers containing random indexes. So that only sequences corresponding to
unique ribosome protected fragments are counted. The reviewer understands that doing
ribosome profiling this way would require a new experiment and he does not expect the
authors doing it for this work, but the issue needs to be at least caveated.

---

## [Author Response]

There seemed to be two general concerns, and some specific concerns. The two general
concerns were, first, that there were not enough control experiments validating the
“Ribosome Residence Time” method; and second, that we did not deal with a
relevant, recently‐published paper by Lareau et al.

With regard to the manuscript’s lack of control experiments validating the
approach, we agree. When we first conceived the approach, we tested and validated it
several ways, but then, when it became clear that the method worked, we lost interest in
validation, moved on to getting results, and neglected to put the validation experiments
into the manuscript. This was a mistake. Readers need to know what the evidence is that
the approach really works. Reviewer 3 asked for some specific experiments, which we had
previously done, but didn’t show. We now have a new Figure (new Figure 2) with four panels, and a new Table (Table 1) showing some of our validation
experiments, and these include most of what reviewer 3 requested. We think these
experiments are very convincing. Inclusion of this validation material makes the
manuscript somewhat longer. If this is a critical concern, this material could be moved
to the supplement.

The second major concern was that we did not deal with a highly‐relevant paper,
Lareau et al., May, 2014. We had finished writing our manuscript in April, before the
Lareau paper came out. For various reasons, we did not submit the manuscript to
*eLife* until June, and in the meantime we did not follow the relevant
literature as closely as we should have. We were not aware of the Lareau paper until the
reviewers pointed it out. Of course, it is a highly relevant paper that needs to be
addressed. We have addressed it in this revised manuscript in a fair amount of detail.
The biggest impact is that the Lareau paper provides us with short footprints from
anisomycin arrest, and we have applied our method to these short footprints. The new
results obtained are in our opinion really interesting, and they are described in a new
section of text, a new Figure, and a new Table. We agree with Lareau et al that these
short footprints are reporting on a different translational event than the long
footprints, so the analysis of the short footprints does not in any way conflict with
any of our previous conclusions, but does give us significant new conclusions. We also
think it is striking that, although we think the Lareau et al. analysis was very good,
nevertheless our analysis got quite a bit more out of the short footprint data, showing
the value of our new method. Finally, Lareau et al. got some long footprints without use
of cycloheximide, and when we analyzed these by our methods, we got a reasonable
correlation (0.47) with our results, which argue that cycloheximide is not introducing
severe artefacts.

*1) The authors suggest that AT rich codons are decoded more rapidly than GC rich
codons but this is not clearly shown in the manuscript. At first this seemed
counterintuitive but the authors suggest an interesting possibility that for GC rich
codons incorrect tRNAs might dwell in the A site longer than they would at weaker
codons and the aggregate of those non-productive interactions might increase the step
time at these codons. The authors do not cite any biochemical studies that would
support this conclusion and in fact they do not cite previous biochemical studies in
the manuscript in many places where that would be appropriate*.

We have cited more biochemical studies, including four relevant to the AT vs GC codon
issue. But we are hesitant to go too far down this path. There have been many
biochemical studies, and they have come to all sorts of conclusions, often conflicting,
and in some cases we are just unable to evaluate these studies. We cannot cite them all,
and we are uncomfortable with picking through them and citing the ones that are, in
hindsight, compatible with our conclusions. So, we are citing a few, including ones we
think are highly relevant, and reviews.

We have also added further information on the relative RRTs of GC vs AT rich codons, and
done a statistical test, which indeed shows a significant difference between the
AT‐rich codons and the GC‐rich codons (p < 0.003).

*2) The manuscript at many places simply states conclusions without providing any
reference (where that would be appropriate) or argument for the conclusion.
Substantiation of the conclusions must be included*.

No examples were given, so we are not sure exactly what conclusions are being referred
to. We have gone through the manuscript looking for such cases, and have tried hard to
add citations, or otherwise provide a reason. If the revised manuscript still has this
defect, we would be happy to provide further citations if the reviewers will point out
the relevant statements.

*3) The major critique of the more critical reviewer is as follows: “Two
of the datasets used in the manuscript were previously published by the same group
and two have not been published before, however the authors state that the other two
were also obtained ”for other reasons in the Futcher lab“. Therefore
while the manuscript provides some newly generated experimental data, the primary
focus is on computational analysis and in particular on RRT. How valuable and useful
is RRT? This, unfortunately, is not very clear. The very fact that it correlates with
codon usage merits further investigation as it may indeed provide potentially useful
characteristic for decoding rates distributions of the codons with the same sequence.
Unfortunately RRT is poorly characterized in the manuscript. Its relationship to the
decoding is not explored beyond correlations with codon usage. Also the manuscript
misleadingly treats RRT as a measure (rather than potentially related statistics) of
codon decoding rates*.*”*

Yes. As we said above, we had originally done validation experiments, but (wrongly)
neglected to put them in the manuscript. The new Figure 2 now shows the results of four of our validation experiments. There is a
positive and a negative control with simulated data; of these, the negative control with
simulated data is the experiment reviewer 3 requested. There is also a positive and a
negative control with real data (positive was a serine starvation experiment; negative
was the RRT analysis of RNA seq data (i.e., 30 bp RNA fragments, but no footprinting).
These two were also essentially experiments the reviewer asked for. The reviewer also
asked about analysis of the histidine starvation data in the Lareau paper as another
positive control, and we analyzed these data and give the result in the text. We will
not go through these experiments in detail here, as we hope they are clear in the
revised manuscript. Additional points:

*4) The title “Measurement of decoding rates of all individual codons in
vivo” is highly misleading*.

*Lys codon AAA at a position i of mRNA x is not the same as Lys codon AAA at a
position j of mRNA y. AAAix and AAAjy likely would have different decoding rates. It
would be wonderful to be able to measure decoding rates of individual codons, but
this is not the case here. RRT is a relative footprint density of individual codons
within a window of 10 neighbor codons averaged over all codons that share the same
sequence*.

We have changed the title.

*5) Before averaging, it makes sense to show that the average is a useful
characteristic of the distribution. It would be the case if the distributions were
normal or at least could be approximated to normal, as this might not be the
case*.

*If the distributions are not normal, the authors may explore other descriptive
statistics of the distribution (for example median) for their relationship with codon
usage, tRNA gene copy number or tAI. But it is highly important to obtain descriptive
statistics of the distributions first. Even if the distributions are normal, the
dispersion of these distributions may not be the same*.

Reviewer 3 asks about the distributions of the window frequencies, and whether the mean
is a good summary statistic or not. This was an interesting question. We have now
provided information on this point in Materials and methods. To summarize, the
distributions are not normal. But the reason is, perhaps, innocuous: we required windows
that have at least 20 reads, and three non‐zero positions. But still, quite
often, a frequency at a particular position in a particular window is zero, because
there are no reads at that position, and so the distributions often have a mode at zero.
We think that as the number of reads goes up and up, and the number of positions with
zero reads goes down, the distributions would approach normal. The distribution is not
one that makes the mean an inappropriate statistic.

In any case, we explored summary statistics other than the mean. We repeated all the
analysis using the median instead of the mean. As now reported in Materials and Methods,
the Spearman rank correlation between the results using the median and the results using
the mean is 0.97. That is, results are essentially identical. We had a conversation
amongst ourselves as to whether the mean was slightly better, or the median was slightly
better, but with a correlation of 0.97 it didn’t really matter, and the mean
contains slightly more information. So we have stayed with the mean. We feel that when
two summary statistics, the mean and the median, give the same answer, the results must
be robust.

*6) The procedure for RRT may have a hidden relationship with the codon usage,
thus explaining the observed correlation. This could be the case because of intrinsic
non-randomness of codon sequences, which effect codon distributions within the
10-codon window. Say a codon X may appear more frequently in the windows centered
around the codon Y than in the windows centered around the codon Z. To explore
whether there is a hidden relationship to codon usage, the authors should assign
experimental footprint densities to codons randomly. Then calculate RRTs and explore
how obtained RRT values relate to the codon usage*.

*In addition to that it would make sense to carry out RRT calculations for naked
mRNA controls and explore obtained distributions in a similar manner*.

We went through the negative control procedure the reviewer suggests, and there is no
signal; all the RRT values come out to essentially 1. This is what is shown in Figure 2. But also, this procedure is implicit in
our method for calculating p‐values for the RRT scores (Table 2, Methods and Materials). That is, the small p‐values
imply that in the randomization experiment, there is not any strong signal. Also, as
suggested, we did the RRT calculations for naked mRNA controls, and again there is no
signal (except at the termini due to enzyme base specificity); this is shown in Figure 2.

*7) Does RRT really relate to codon decoding rates? Intuitively it should, but
this needs to be shown. We know from a number of experiments, including ribosome
profiling that concentration of tRNAs effect the decoding rates. There are publicly
available datasets where particular aminoacylated tRNAs were depleted, see for
example*
[34]
*eLife [PMID: 24842990] for S. cerevisiae or*
[35]
*Nature [PMID:22456704] for bacterial organisms. There might be more. The authors
should calculate RRT for these datasets and characterize RRTs between these
datasets*.

The reviewer suggests we look at the dataset for serine‐starved *E.
coli*. We had previously done this but not shown it; it was one of our first
tests of the method. The RRT analysis shows big peaks for the serine codons, and this is
now shown in Figure 2 and the accompanying
Table. We also looked, less successfully, at the Lareau data for histidine starvation.
This is now described in the text. It was less successful because the Lareau dataset for
histidine starvation was actually rather small, too small for RRT analysis as described
here. Nevertheless, with relaxed quality filters, we did see good‐sized peaks for
the two His codons in the 3‐AT treated cultures, but, importantly, not in the
non‐starved cultures. We are not putting the figure in the paper, because it is
not real RRT analysis, because we had to relax the quality filters to get enough
windows.

*8) If RRT is indeed a good statistics it should work well not only in yeast, but
also in other organisms, thus it is important to carry similar analysis on other
publicly available datasets. The authors did that for the very first ribosomal
profiling dataset*
[28]
*dataset, which was generated with cycloheximide pretreatment. It also has a
relatively poor coverage. For yeast it would be advisable to use*
[34]
*Elife [PMID: 24842990] data*.

*RRT should also correlate with other ways of measuring elongation speed. For
example, the authors could calculate RRT for*
[29]
*Cell [PMID: 22056041] and explore whether an aggregated RRTs for the codons of
individual coding sequences could predict differences in the speed of ribosomes over
individual mRNAs estimated with the pulse chase experiment described in this
work*.

Yes, the method works well in other organisms. You now see some evidence of that here.
We have analysed a lot of existing ribosome profiling data for various organisms from
databases. Obviously we cannot put it all in this manuscript. Several quite interesting
things have come out of this analysis, and we are in the early stages of planning
additional manuscripts. However, we did apply the method to the Lareau et al
cycloheximide datasets, and do now report the results here in this manuscript. The
correlations with our results are on the low side; 0.2 to 0.5, but still, they are all
positive correlations, and we strongly believe that they are relatively modest because
Lareau, like Ingolia, add cycloheximide first, then grow the cells a bit, then harvest,
whereas we flash‐freeze first, then add cycloheximide to the frozen cells.

The fact that it works well with data in databases from other organisms is of course a
reason to publish the paper, so that others will have access to these methods and can
also do this analysis.

*9) The critical reviewer is skeptical regarding the use of ribosomal profiling
data for estimating decoding rates for several reasons. The time that ribosome spends
at a particular codon is only one of the factors affecting the number of footprints
aligning to that codon. The others are sequence coverage (i), initiation rate for the
start codon of corresponding ORF (ii) and concentration of corresponding mRNA (iii).
These factors can be estimated (e.g. mRNA-seq can estimate relative mRNA levels) or
the data could be normalized in the ways that would minimize the contribution of
these factors, e.g. a footprint density at a particular location can be normalized
over the cumulative density for the entire dataset or for individual mRNAs. Such
normalization procedures aren't perfect and may generate certain artifacts and
Gardin et al do discuss it rightfully to some extent. However, there are other
factors that are more difficult to take into account, such as the effect of
antibiotics on capture of the ribosomes at particular locations or even at specific
ribosomal conformations leading to differing length of footprints, see*
[34]
*eLife [PMID: 24842990] for more information. The reviewer is also surprised and
disappointed that this very important and highly relevant article is not mentioned in
the manuscript*.

*The other factors effecting densities are those related to the biases of cDNA
library preparation. Their presence can be easily seen in the data analyzed here with
RRT as well, e.g.*
Figure 2*; codon 10
corresponds to the 5' ends of the footprints and show a variability comparable
to that of position 6 (btw plotting 64 curves on the same diagram is not very
effective, the authors should explore other statistics for measuring variability
within a distribution). Most likely this variability is due to sequence specificity
of RNAse cleavage and/or adapter ligation. The other factor that is highly relevant
to measuring decoding rates is PCR amplification of cDNA libraries. PCR amplifies
fragments non-linearly, and the ratio between a low abounded fragment and a highly
abounded fragment would likely increase after PCR. To control for that the initial
step of cDNA amplification should be carried out with RT primers containing random
indexes. So that only sequences corresponding to unique ribosome protected fragments
are counted. The reviewer understands that doing ribosome profiling this way would
require a new experiment and he does not expect the authors doing it for this work,
but the issue needs to be at least caveated*.

The thrust of point 9 seems to be that we should be able to get similar results by other
methods, by estimating the sizes of various effects. Well, maybe. But we are not
pursuing other methods; we are trying to describe this one. Also, as we say at the
beginning of this manuscript, we think the estimates and guesses involved in making the
calculations the reviewer suggests are problematic, and quite likely to lead to the
wrong answer. That our approach by‐passes all this guessing and estimating is a
lot of the point.

The reviewer also mentions the well‐known fact that PCR can create sampling
artefacts. But if any method can defeat PCR sampling problems, it is this one, because
we simply consider each and every window as an independent experiment, no matter the
frequency of reads in that window. Relatively rare PCR sampling artefacts will affect
one window at a time (out of thousands) and so have a negligible impact on our approach.
The fact that we have correlations up to 0.96 between our experiments demonstrates that
random noise such as introduced by PCR sampling artefacts cannot be a big issue.

Yes, we need to cite and write about Lareau et al. We are sorry to have missed this
paper, and thank the reviewers for pointing it out. The revised manuscript talks
extensively about the Lareau et al. results, which are really interesting.